# Performance of Fabrics with 3D-Printed Photosensitive Acrylic Resin on the Surface

**DOI:** 10.3390/polym16040486

**Published:** 2024-02-09

**Authors:** Payton Becker, Izabela Ciesielska-Wrόbel

**Affiliations:** Department of Textiles, Fashion Merchandising and Design, College of Business, University of Rhode Island, 55 Lower College Road, Kingston, RI 02881, USA; p_becker@uri.edu

**Keywords:** 3D printing on textiles, photosensitive acrylic resin, polyjet print on textiles, cotton, polyester, fabric performance, colorfastness to light, washing

## Abstract

Additive manufacturing (AM), also known as three-dimensional printing (3DP), has been widely applied to various fields and industries, including automotive, healthcare, and rapid prototyping. This study evaluates the effects of 3DP on textile properties. The usability of a textile and its durability are determined by its strength, washability, colorfastness to light, and abrasion resistance, among other traits, which may be impacted by the application of 3DP on the fabric’s surface. This study examines the application of photosensitive acrylic resin on two fabric substrates: 100% cotton and 100% polyester white woven fabrics made of yarns with staple fibers. A simple alphanumeric text was translated into braille and the braille dots were 3D printed onto both fabrics. The color of the printed photosensitive acrylic resin was black, and it was an equal mixture of VeroCyanV, VeroYellowV, and VeroMagentaV. The 3D-printed design was the same on both fabrics and was composed of braille dots with a domed top. Both of the 3DP fabrics passed the colorfastness to washing test with no transfer or color change, but 3D prints on both fabrics showed significant color change during the colorfastness to light test. The tensile strength tests indicated an overall reduction in strength and elongation when the fabrics had 3DP on their surface. An abrasion resistance test revealed that the resin had a stronger adhesion to the cotton than to the polyester, but both resins were removed from the fabric with the abrader. These findings suggest that while 3DP on textiles offers unique possibilities for customization and design, mechanical properties and color stability trade-offs need to be considered. Further evaluation of textiles and 3D prints of textiles and their performance in areas such as colorfastness and durability are warranted to harness the full potential of this technology in the fashion and textile industry.

## 1. Introduction

Additive manufacturing (AM), also known as three-dimensional printing (3DP), was defined by the International Organization for Standardization (ISO) and the American Society for Testing and Materials (ASTM) as processes of joining materials, usually layer upon layer, to make parts according to a computer-aided design (or 3D model data) [1]. In these processes, geometrical representation creates physical objects by successive addition of material [2,3]. This type of production allows the creation of complex objects, often with ultrafine details, surpassing other less sophisticated production techniques, such as injection molding. There has been a significant increase in the use of 3DP in different areas, such as printed parts or entire customized medical implants [4,5], parts for aerospace devices [6], dental braces [7], composites [8], lattice structures [9], and parts for the automotive industry with improved thermomechanical properties [10]. Also, due to the growing popularity of AM, other nontraditional areas, such as textiles and the fashion industry [11,12], are now gaining momentum. Research and industrial applications in these areas encompass the printing of entire pieces of fabric [13] or apparel [14], printing parts of apparel [11], including decorations [11], as well as printing onto fabrics [15]. A fully printed garment is often produced without incorporating details of the fabric substrate, such as fibers and yarn specifics, resulting in the apparel piece resembling more of a shell.

### 1.1. Polyjet Printing Directly on Textiles

Polyjet 3DP directly on textiles is classified as a direct, liquid-based, polymerization-based printing technique [2]. Polyjet technology was introduced to the market by Objet Geometries, Ltd. in 2000. However, it was Stratasys, Ltd. that further developed this printing technique and introduced a 3D polyjet printer suitable for printing on textiles in 2019 [2,16]. In the process of 3DP on textiles, a fabric is placed on the printer table. Next, the droplets of a photosensitive acrylic resin are jetted onto the surface of the fabric according to a design and builds the design layer by layer. An ultraviolet (UV) light cures each layer. This is possible because the photosensitive acrylic resin contains compounds, usually monomers or oligomers in a liquid form, that, when subject to a source of light of a particular wavelength, get cured; thus, their shape is fixed. This process is also called photocuring [17]. Polyjet printers often use UV-visible light to cure the resin and produce solid structures. Figure 1 presents a scheme of 3D polyjet printing on fabrics. When a single layer of the resin hardens after being cured with UV light, the next layer is deposited on the cured resin or aside from it, depending on the design. While the minimum thickness of a print on the fabric is 0.0079 inches (0.2 mm), it is possible to print relatively tall objects on the surface of fabrics, up to 7.9 inches (20 cm) [16], which is related to a space limitation in the printer and is unrelated to the technology itself. Typically, curing takes place in the range of 315–405 nm of UV light [17,18], but the curing wavelength depends on the parameters of the resin to be cured [19].

Since 3D printing on fabrics is a relatively new technology introduced to the market by Stratasys in 2019, there is no sufficient information on the physical performance of fabrics with a 3D print. Although there is some information available on the adhesion of the resin to fabrics [15,20,21], it is unclear how fabric with an attached resin can perform and how resilient the resin and fabric are after the fabric is subjected to different tests simulating regular textile usage.

### 1.2. Aim of This Study

This study aims to learn about the performance of textiles with 3DP on their surface. It is an initial, exploratory step toward understanding the potential impact of 3DP on textiles. The usability of textiles and their durability are determined by their strength, washability, and colorfastness to light, among others, which may be impacted by the application of 3DP on the fabric’s surface.

## 2. Materials and Methods

### 2.1. Fabrics

Two types of fabric substrates were used to test the impact of 3DP on their surface: 100% polyester and 100% cotton white woven fabrics. These materials were chosen due to commonly being used in the fashion industry. The 100% staple fiber polyester fabric was sourced from Artex International, Pakistan, and the 100% staple fiber cotton fabric was sourced from Test Fabrics, Inc., West Pittston, PA, USA. The fabric substrates’ characteristics were measured according to the standards of the American Society for Testing and Materials (ASTM): for fabric weight [22], thickness [23], and fabric count [24] and are presented in Table 1.

### 2.2. The Design for 3DP on Textiles

The design utilized to print on the fabrics was created by the authors as a part of research on the preferences of people with visual impairments and blindness for clothing information written in braille, which is an independent study from the currently reported research. The design used texts about textiles, which were translated into braille. These texts were selected as examples of the most helpful information for visually impaired and blind individuals when learning about the textiles that they interact with. Other reasons why the braille designs were chosen to test the performance of the textiles include regularity in the printed resin’s color, shape, and size. The regular spacing between the braille dots allows for easy access to the textile surface for observations.

Braille is a system of touch reading and writing for visually impaired and blind individuals in which raised dots represent the letters of the alphabet [25]. The following text was translated into braille: “white”, “small”, “100% cotton”, “wash cold”, and “tumble dry low”, which is an example of concise information about clothing that may be printed on fabric or a clothing tag for visually impaired and blind individuals. “White” refers to the color of the fabric; “small” refers to the size of the clothing; “100% cotton” refers to the raw material that the fabric is made of; “Wash cold” indicates that the fabric should be washed in lower temperatures, usually not more than 30 degrees Celsius (86 degrees Fahrenheit); and “Tumble dry low” means that the material should be dried in the dryer on a low heat setting, usually not more than 50 degrees Celsius (around 122 degrees Fahrenheit). The authors used a free translator at https://www.brailletranslator.org/ accessed on 1 June 2023) to translate the alphanumeric texts. Figure 2 shows the source alphanumeric text (left) and its braille translation (right).

The braille created at https://www.brailletranslator.org/ (accessed on 1 June 2023) was copied and pasted into Linearity Curve, a vector creation software version 5.3.3., and converted into 3D braille design using the “Swell Braille” font that adheres to the spacing guidance provided by the Americans with Disabilities Act (ADA) [26]. After creating vector outlines from the braille text, the file was exported as a Scalable Vector Graphic (.svg) file and then imported to AutoCAD to set the dimensionality of the dots according to the ADA guidelines [26]. The three-dimensional design was then exported as an OBJ (.obj) file containing the three-dimensional information needed for 3DP. The OBJ file was shared with Stratasys, Ltd. who completed the printing but scaled down the design by 11% for the test samples used in the current study, meaning that the samples for this study are no longer ADA-compliant, which includes the distances between the dots within symbols and between symbols, and their original dimensions, which were 0.06 inches (1.54 mm) for a diameter of dots and 0.019 inches (0.48 mm) for the height of the braille dots. The scaling down was dictated by the need to fit as many printouts as possible on the supplied fabrics while economizing on fabrics and resin.

### 2.3. The Resin Used in 3DP on Textiles. Color of the Resin—Black Made of Cyan, Magenta, and Yellow

The resin used to produce the black dots on the textile substrates is an equal mixture of Stratasys’ transparent VeroVivid™ resins: VeroCyanV (RGD845) [27], VeroYellowV (RGD838) [28], and VeroMagentaV (RGD852) [29]. The VeroVivid™ resins are acrylic photopolymers, but most of their chemical components are proprietary [16]. These resins have a reported tensile strength of 7250–9450 psi, a flexural strength of 11,000–16,000 psi, and a heat deflection temperature of 40–50 °C [16]. The Stratasys J850™ Tech Style™ printer was used to print on the fabric substrates. 

The safety data sheets of all three acrylic printable inks, VeroCyanV (RGD845) [27], VeroYellowV (RGD838) [28], and VeroMagentaV (RGD852) [29] are similar. They list 15 ingredients for VeroCyanV and VeroMagentaV, and 16 ingredients for VeroYellowV. Eight of these ingredients and their contribution to the mixtures of VeroMagentaV and VeroCyanV are not revealed because they are proprietary information. The VeroYellowV mixture contains nine ingredients with proprietary information. The exact contribution of these proprietary ingredients to the mixtures was not revealed as these are trade secrets. They contribute to the mixture between 0.1 and 30% of the weight of the mixture. The proportion in the mixture of the remaining nonproprietary ingredients is 1–3% of the weight for caprolactone acrylate and 0.1–0.3% for acrylic acid, 2-hydroxyethyl ester; 2-propenoic acid, 1,2-ethanediyl ester; 2,6-bis(1,1-dimethylethyl)-4-methyl-phenol; camphene; glycerol, propoxylated, esters with acrylic acid; and acrylic acid. 

Caprolactone acrylate is a derivative of caprolactone (or ε-caprolactone). This substance contains acrylate groups, which makes it photopolymerizable. Other researchers reported that typical acrylate photoresins consist of functional monomers, oligomers, a photoinitiator, and an optical absorber [30,31,32]. The modification of caprolactone with acrylate allows it to be used in applications like 3D printing resins where light-induced curing is required.

2-hydroxyethyl (HEA) ester with acrylic acid [33] is a comonomer of HEA and acrylic acid, which is used in making polymers, especially as a reactive resin component due to its ability to rapidly polymerize under UV light. It contributes to the formulation of photopolymers that are curable and can provide desired mechanical properties and surface finish to the printed objects. It may have been added to the mixture of the resin to enhance the print’s durability, flexibility, and adherence properties. This monomer can be used in various industrial applications for its desirable properties, such as in coatings, adhesives, and resins for 3D printing [34]. 

2-propenoic acid, 1,2-ethanediyl ester [35] may have been used in 3D polyjet printing as a component in photopolymer resins due to its ability to form cross-linked structures upon exposure to UV light. This cross-linking enhances the mechanical strength and durability of the printed objects. It is often used to improve the properties of the final printed product, such as hardness, toughness, and solvent resistance. 

2,6-bis(1,1-dimethylethyl)-4-methyl-phenol [36] is used as an antioxidant and preservative. It may have been used in the resin to prevent its aging and extend its shelf life or the cartridges containing these resins. 

Camphane [37] may have been used in the mixture as an odor neutralizer for other ingredients used in the mixtures of acrylic inks. However, it is also known to be used as a porogen, which is a substance added to a material to create pore structures. In the context of photosensitive resins for 3D printing, camphene could serve several purposes. For instance, it could have been used in the mixtures to control the mechanical properties of the mixture [38]. The introduction of pores can alter the mechanical properties of the printed object, potentially making it lighter or more flexible, depending on the specific requirements of the application. Adding camphene may influence the curing process of the resin. By affecting how the resin reacts to light or heat during the printing process, it can improve the overall quality and stability of the final product. Camphene may help achieve finer details and smoother surfaces in the printed objects, a crucial aspect of high-quality 3D printing.

Propoxylated, glycerol, esters with acrylic acid [39] were used in photosensitive acrylic resins for 3D printing due to their ability to enhance photosensitivity, control viscosity, improve curing properties, and ensure mechanical durability of the printed objects. They also allow for compatibility with various additives and exhibit low shrinkage during curing, therefore maintaining the dimensional accuracy of the 3D-printed items [40]. 

Acrylic acid [41] is an organic substance, a byproduct in the production of ethylene and gasoline. It may have been added to the mixture due to its high reactivity to light, especially UV light. It could have helped to control exposure to light when cured and solidified each layer of the resin precisely. However, the curing process of a resin with acrylic acid is fast, which makes this substance an essential ingredient for the speed and efficiency of the 3D printing process. Acrylic acid contributes to the adhesive properties and overall mechanical strength of the final printed material. This is important for creating durable and functional 3D-printed objects [42].

### 2.4. Colorfastness to Washing

The American Association of Textile Colorists and Chemists (AATCC) test method 61-2013e2(2020) for Colorfastness to Accelerated Laundering [43] was used to evaluate the colorfastness of the samples with 3DP on its surface in comparison to the unlaundered material and the reference material, which did not have any resin applied to its surface. Colorfastness is the resistance of a material to change in any of its color characteristics, to transfer of its colorant(s) to adjacent materials, or both, as a result of the exposure of the material to any environment encountered during the test [43]. This test method utilizes an accelerated washing that simulates five home launderings at a time. The system used in our lab is an Atlas Launder-Ometer. As per the AATCC procedure, the test samples were cut to 50 by 150 mm and stapled to multifiber adjacent fabric for the staining test. The multifiber adjacent fabrics are woven and have six reference stripes made of different textile raw materials: acetate, cotton, nylon, polyester, acrylic, and wool. They are attached to tested fabrics to see if and how the multifiber stripes are affected by a tested fabric. One polyester reference sample and three polyester samples with 3DP on their surface were tested, while only one cotton sample and one cotton reference sample were able to be tested due to the limited quantity of the cotton fabric with the print on it. The laundering containers were filled with one sample each, 0.225 g of AATCC standard powder detergent, 150 mL of deionized water, and 50 steel balls. The samples were laundered as required by test conditions 2A at 49 °C for 45 min before being air dried. The first polyester sample went through one cycle, the second went through two cycles, and the third went through three cycles to compare the appearance of 5, 10, and 15 launderings, respectively. The samples were then compared to unlaundered samples and the laundered reference samples, using the Gray Scale for Color Change [44] to evaluate the color change and the Gray Scale for Staining [45] to evaluate the stain. 

### 2.5. Colorfastness to Light

In the colorfastness to light test, fabrics were exposed to UV light to learn if they fade. Both the fabric substrates and the 3DP were examined after the test. AATCC test method 16.3-2020 was used to determine the samples’ colorfastness to UV light [46]. The test method uses a Xenon-Arc to deliver UV to the sample. The Xenon-Arc machine used for this research was the Q-Sun Xenon Test Chamber (Model XE-3-HS). The UV light parameters were as follows: irradiance 1.10 W/m^2^ at the wavelength of 420 nm. As per the AATCC procedure, the test samples were cut to 70 by 120 mm and stapled into the test specimen mask before being set into the frames. One polyester reference and three polyester samples with 3DP on were tested, while only one cotton reference and one cotton sample with 3DP on it were tested. The test samples and the reference samples, which did not have 3D-printed resin, were exposed to 10, 20, and 30 h of light. The exposed samples were compared to the covered portion and the color change was evaluated using the Gray Scale for Color Change (AATCC EP1) [44]. The exposed samples’ color change was also compared to the reference samples.

### 2.6. Tensile Properties of the Fabrics

The ASTM test method was used to evaluate the breaking strength and elongation of the samples [47]. The strip method was used on a constant-rate-of-load machine, Admet and recorded in its software MTestQuattro 2017 (https://www.admet.com/products/controllers-and-indicators/mtestquattro/, accessed on 1 August 2023). As per the ASTM procedure, the test samples were cut to 25 mm wide and were classified as 1C-L [47]. While the procedure called for five specimens to be cut from each direction, the polyester sample yielded six in the warp direction and three in the weft direction alongside three polyester reference samples from both directions. Similarly, three cotton samples in each direction were tested alongside three cotton reference samples. The samples and reference samples were clamped into the tensile testing machine with clamps and the test was run via the software MTestQuattro 2017 (https://www.admet.com/products/controllers-and-indicators/mtestquattro/, accessed on 1 August 2023), which gave the breaking force and elongation results. The clamps’ capacity was 500 lbf and they were manually tightened.

### 2.7. Abrasion Resistance

The ASTM test method to measure fabric pile resistance to abrasion [48] was modified to determine the resin and fabric’s ability to withstand abrasion. As per the ASTM procedure, 135 mm circles with a less than 6.5 mm hole were cut from the fabric [48]. In a deviation from the procedure, only the fronts of the fabrics were tested, so two circles were cut from the polyester sample and one circle was cut from the cotton sample. One reference sample was cut for both the polyester and cotton for comparison. Option B, Light Duty Procedure, was followed, excluding testing the samples’ back. As per the procedure, the samples were tested on the Taber rotary platform abrader with CS-10 wheels and a loading mass of 250 g/wheel for 300 cycles. Instead of comparing the samples to the ASTM Photographic Scale for Pile Retention, the samples were examined for the overall appearance of the resin dots (i.e., the shape, condition, etc.) and compared to both the original dots and the state of the reference samples.

## 3. Results

### 3.1. Tests Results of Colorfastness to Washing 

Both the polyester and cotton samples passed the AATCC colorfastness to washing and staining tests with ratings of 5, which indicates no change in color [44,45]. The polyester sample remained unchanged even after three accelerated washings, equivalent to 15 home washings (see Figure 3). While there were no colorfastness issues for the cotton sample, we observed puckering around the resin dots and a general shrinkage of the design (see Figure 4 and Figure 5). The resin did not stain the substrate fabrics, nor the reference fabrics. While measuring the dimensional change of the textile substrates was not considered in this study, we noticed that the cotton did shrink during its laundering, both in warp and in the weft direction, as presented in Figure 5. After one accelerated wash, the observed shrinkage was 3.6% in the warp direction and 7.4%. in the weft direction. The shape and the size of the dots were not affected by the washing process.

### 3.2. Tests Results of Colorfastness to Light 

The resin dots on both the polyester and cotton samples showed a significant color change from black to dark blue after 10 h of light exposure, scoring 3 on the Gray Scale for Color Change [44]. The color changed to a lighter blue after an additional 10 h of light exposure for a total of 20 h, scoring 2 on the Gray Scale for Color Change. There was no observed change after another 10 h for a total of 30 h. Thus, the dots after 20 and 30 h of exposure had a similar light blue color while the dots exposed to 10 h were less affected. The fabric substrate itself was unaffected by the light. Figure 6 shows the polyester samples alongside the reference fabric, Figure 7 shows the cotton sample alongside the sample fabric, and Figure 8 shows a close up of the resin dots on polyester before and after 10, 20, and 30 h of exposure to UV light.

### 3.3. Tests Results of Tensile Properties 

Table 2 and Table 3 present the results of the tensile tests performed on polyester and cotton fabrics with and without the resin print. Fabrics were tested in the warp and weft directions. Fabric specimens after the test are presented in Figure 9 (polyester), and Figure 10 (cotton). While reference fabrics, in general, were torn in the center, fabrics with the print were torn close to the clamps or the tensile test apparatus. Due to the presence of the dots on the fabric, the specimens were squeezed between the clamps more than the reference fabrics. If not, the samples would slip out. The crushed dots are observed in Figure 11.

In tests conducted on polyester samples in both the warp and weft directions, those with 3DP exhibited lower breaking force and elongation values compared to reference polyester fabrics without 3DP. Specifically, the breaking force decreased by 28.43% in the warp direction and by 35.36% in the weft direction. Similarly, elongation was reduced by 15.20% in the warp direction and by 27.28% in the weft direction. 

In tests conducted on cotton samples in both the warp and weft directions, those with 3DP exhibited lower breaking force and elongation values compared to reference cotton fabrics, without 3DP. The breaking force decreased by 45.83% in the warp direction and by 44.98% in the weft direction. Similarly, elongation was reduced by 22.96% in the warp direction and by 21.90% in the weft direction.

### 3.4. Tests Results of Abrasion Resistance

The pile abrasion resulted in the removal of the resin dots to different degrees on both the polyester (Figure 12) and cotton samples (Figure 13). For the polyester samples, there were no resin dots left after 80 cycles for sample 1 and no resin dots left after 100 cycles for sample 2. The surface of the polyester samples after the abrasion test were slightly fuzzy in appearance, which is similar to the appearance of the abrasion test on the polyester reference. The polyester dots that were removed by the abrasion test were retrieved and observed. They had some attached fibers that were removed alongside the resin dot (see Figure 14). The single dot presented in Figure 14 has some visible striations on its surface, most probably coming from the deposition of the resin during the printing process. The rough bottom of the dot shows the trace of the fabrics structure to which it was attached. Fractions of the fibers were observed at the bottom of the dot. The average weight of a single dot after it was removed from the fabric was 0.0019 g ± 0.00085. For the cotton samples, only some of the resin dots were removed after the full 300 cycles, while others were crushed, damaged, or torn with the fabric (see Figure 15). Like the polyester sample, the cotton’s surface after the abrasion test was slightly fuzzy in appearance, matching the cotton reference.

## 4. Discussion

### 4.1. Colorfastness to Washing 

The resin dots withstood accelerated laundering without color loss or transfer on both the cotton and polyester substrates. The resin dots also showed no signs of degradation when observed with the naked eye. This means that fabrics with 3DP elements should be safe to launder for up to 15 home washes, an equivalent of three accelerated washes. The shrinkage seen on the cotton sample suggests that the cotton substrate should either be treated with a shrinkage-proof finish or washed before resin application to avoid distortion of the 3DP patterns. It is unclear how the printed dots impacted the shrinkage of the cotton fabric.

### 4.2. Colorfastness to Light

The failure to resist color change in UV light is an area of concern for the 3DP fabrics moving forward. As previously mentioned, the resin dots achieved their black color through a combination of three colored resins: VeroCyanV (RGD845), VeroYellowV (RGD838), and VeroMagentaV (RGD852). The color change of the dots from black to blue suggests that VeroMagentaV and VeroYellowV were not colorfast in UV light, which left behind the VeroCyanV, giving a blue color to the dots. It is unclear if this phenomenon is due to mixing the three colors to obtain black or if it has roots in the ingredients of the VeroCyanV being more resistant to changes when exposed to UV light of 400 nm for at least 20 h. An unmixed black resin is available and may yield different results when exposed to UV light. 

### 4.3. Tensile Properties

The tensile properties tests measured the breaking force and elongation of both the fabric references and samples with 3DP. The results showed that the presence of resin dots had a noticeable impact on the tensile properties of the fabrics. The breaking force and elongation were reduced in the samples with 3DP compared to the reference fabrics. This suggests that the addition of resin affects the mechanical properties of the textiles, potentially making them less stretchable and weaker. This finding is congruent with a general concept observed in textiles that have a coating or a finish on their surface such as resin [49,50]. The resins can restrict the movement of the fiber and yarns in the fabrics, which reduces their flexibility and, in turn, can make the fabrics more prone to tearing or breaking under stress. The 3D-printed resin restricts the movements of fibers because resin acts as a glue, surrounding fibers, attaching to their surface, and bonding neighboring fibers; this effectively reduces their range of movement when subjected to stress.

The reference polyester fabric exhibited a breaking force of 147.11 lbs, while the polyester warp sample with 3DP had a lower breaking force of 105.29 lbs. This indicates a reduction in tensile strength when 3DP is applied to the fabric in the warp direction. The reference fabric had an elongation of 54.94%, while the sample of polyester fabric tested direction showed a slightly lower elongation of 46.59%. This suggests that the presence of 3DP resin reduced the fabric’s ability to stretch before breaking. In the polyester warp sample, there were an average of 154 resin dots present.

The reference polyester fabric in the weft direction had a breaking force of 118.28 lbs, while the polyester weft sample with 3DP had a lower breaking force of 79.46 lbs. Similar to the warp direction, the presence of 3DP reduced the tensile strength of the fabric in the weft direction. The reference fabric had an elongation of 71.35%, whereas the polyester weft sample showed a slightly lower elongation of 51.88%. Again, this indicates that 3DP on the fabric reduced its ability to stretch. In the polyester weft sample, there were an average of 158 resin dots present.

The reference cotton fabric in the warp direction had a breaking force of 41.04 lbs, while the cotton warp sample with 3DP had a significantly lower breaking force of 22.23 lbs. This shows a substantial reduction in tensile strength when 3DP is applied to cotton in the warp direction. The reference fabric had an elongation of 15.29%, whereas the cotton warp sample exhibited an even lower elongation of 11.78%. This suggests that 3DP negatively impacted the fabric’s ability to stretch before breaking. In the cotton warp sample, there were an average of 134 resin dots present.

The reference cotton fabric in the weft direction had a breaking force of 36.57 lbs, while the cotton weft sample with 3DP had a breaking force of 20.12 lbs. Similar to the warp direction, the presence of 3DP reduced the tensile strength of cotton in the weft direction. The reference fabric had an elongation of 39.77%, while the cotton weft sample had a slightly lower elongation of 31.06%. This again indicates that 3DP negatively affected the fabric’s ability to stretch. In the cotton weft sample, there were an average of 127 resin dots present.

The tensile test results demonstrate a consistent trend across all samples and directions. The application of 3DP resin significantly reduced the breaking force (tensile strength) of both polyester and cotton fabrics. Additionally, the presence of 3DP resin led to a decrease in fabric elongation, indicating that the fabrics became less stretchable with 3DP. These findings highlight that 3DP on textiles can compromise the mechanical properties of the fabrics, making them less robust and less capable of withstanding tensile forces. Designers and manufacturers need to consider these effects when incorporating 3DP into textiles, particularly if the fabric needs to maintain its original tensile properties for specific applications. Further research may be necessary to explore methods for strengthening fabrics with 3DP or developing specialized applications where reduced tensile strength is acceptable.

### 4.4. Abrasion Resistance

In addition to the tensile tests, the abrasion resistance tests revealed that the resin dots on both the polyester and cotton samples were susceptible to removal during testing. For the polyester samples, the resin dots were entirely removed after 80 to 100 cycles of abrasion. This suggests a weak connection between the polyester substrate and the resin dots. The fabric surface of the polyester samples became slightly fuzzy, resembling the appearance of the reference fabric after abrasion. The cotton sample’s resin dots withstood the abrasion better than those on the polyester. Some resin dots remained on the cotton sample after the standard 300 abrasion cycles, but there was evidence of tearing around the resin dots. This suggests a better bond between the cotton substrate and the resin dots than the one between the polyester and resin, since they were more easily removed. These results suggest that 3DP on textiles may not be highly durable when subjected to abrasion. 

## 5. Summary

This study provides valuable insights into the performance of textiles with 3DP on their surface. The colorfastness to washing was generally satisfactory, but the colorfastness to light indicated that the resin dots may undergo color changes with prolonged UV exposure. Tensile properties were adversely affected by the presence of resin, making the fabrics less stretchable and weaker. Abrasion resistance tests revealed that the resin dots were susceptible to removal during abrasion.

Additionally, due to the nature of the cured resin and the 3DP design studied in this research, it is not recommended to wear fabrics with such a design in direct contact with the skin.

These findings suggest that while 3DP on textiles offers unique possibilities for customization and design, there are trade-offs in terms of mechanical properties and color stability that need to be considered. Further research and development in areas such as colorfastness, mechanical reinforcement, and durability of 3DP on textiles are warranted to harness the full potential of this technology in the fashion and textile industry.

## 6. Conclusions

Woven cotton and polyester fabrics with 3D-printed braille using photosensitive acrylic resin passed colorfastness to washing tests. The fabrics remained unstained by the resin even after three accelerated washing tests, and the resin’s appearance did not change postwashing. However, the cotton fabric experienced shrinkage.The 3D-printed black resin on the fabrics significantly faded and turned blue after exposure to as little as 20 h of UV light. This change appears to be related to the resin’s composition. Identifying the specific resin ingredient responsible for this is challenging due to the proprietary nature of the resin components.Tensile tests revealed that fabrics with 3D-printed braille, which creates periodic effects on the fabric’s surface, are significantly weaker. In general, periodic 3D printing on fabrics, such as braille, diminishes the strength and elongation of the materials.Abrasion resistance tests showed that braille dots could be relatively easily removed from the fabric surfaces during friction. However, they adhered better to cotton fabric due to its hairiness. While cotton fabric tends to retain damaged or smashed braille dots, they tend to chip off the polyester fabric.

## 7. Future Research

This study opens up avenues for future research, including the evaluation of the thermophysiological comfort of fabrics with different types of 3DP on clothing. While 3D-printed dots may not have a significant impact on sweat dissipation from the skin through a textile substrate, larger 3D-printed decorative designs may prevent sweat from evaporating. Investigating how 3DP on textiles performs when exposed to perspiration (colorfastness to sweat), another common scenario in clothing usage, is worth investigating. This test can be accompanied by moisture management analysis of fabrics. Another test that can be performed on fabrics with 3DP is assessing the resilience of 3DP on textiles when subjected to ironing, a typical garment care practice. While we made some observations during the current study—wrinkling of a textile substrate after the washing process, which may impact the appearance of the printed design and its functionality—it was not the focus of this study; however, the avenues to handle textiles and clothing after the washing process is an avenue worth investigating. Additionally, expanding the scope of designs beyond dots and a singular, mixed color to examine the impact of various patterns and shapes on fabric performance is one of the most important future tests from the standpoint of textiles’ mechanical properties. Continued research in these areas will contribute to a better understanding of the capabilities and limitations of 3DP on textiles, facilitating its integration into the fashion and textile industry.

## Figures and Tables

**Figure 1 polymers-16-00486-f001:**
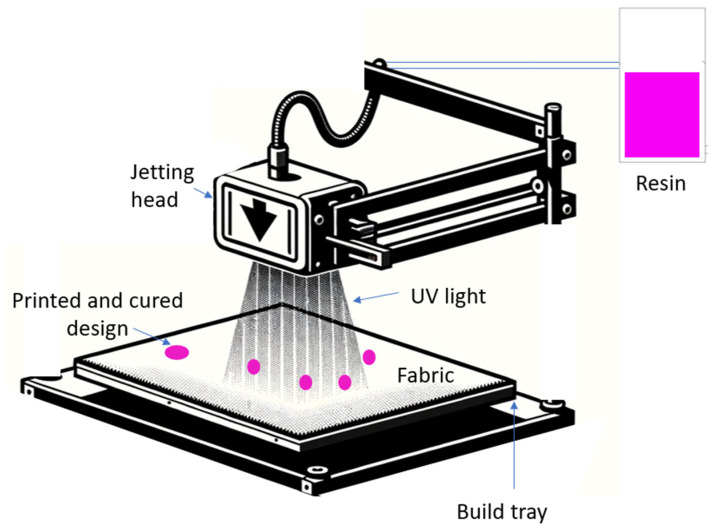
Schematic of 3D polyjet printing on textiles. Fabric is placed on a build tray and the jetting head releases a photosensitive polymeric resin that is subsequently cured by UV light (authors’ image).

**Figure 2 polymers-16-00486-f002:**
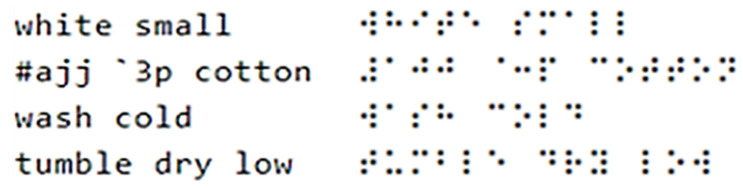
An alphanumeric text (**left**) contains the symbol #ajj `3p, which is identifiable by braille as 100%. The braille translation (**right**) of the alphanumeric text was provided by https://www.brailletranslator.org/, accessed on 1 June 2023.

**Figure 3 polymers-16-00486-f003:**
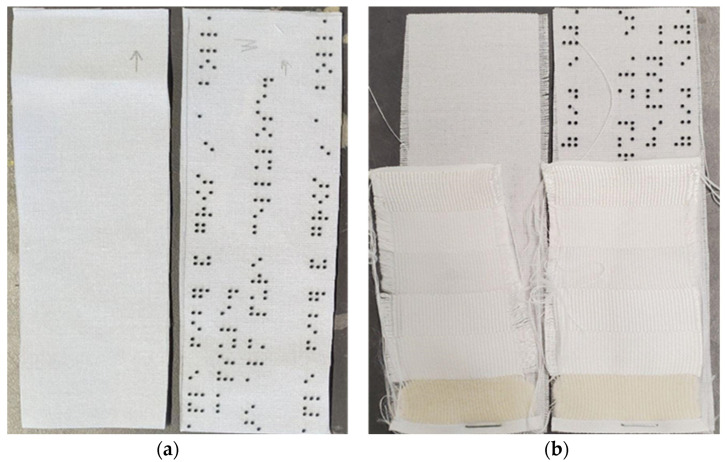
The polyester reference and sample: (**a**) before accelerated washing; (**b**) after accelerated washing.

**Figure 4 polymers-16-00486-f004:**
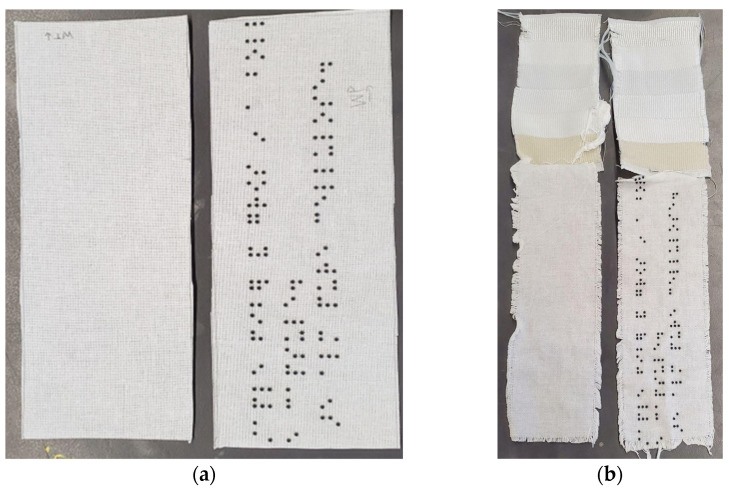
The cotton reference and sample: (**a**) before accelerated washing; (**b**) after accelerated washing.

**Figure 5 polymers-16-00486-f005:**
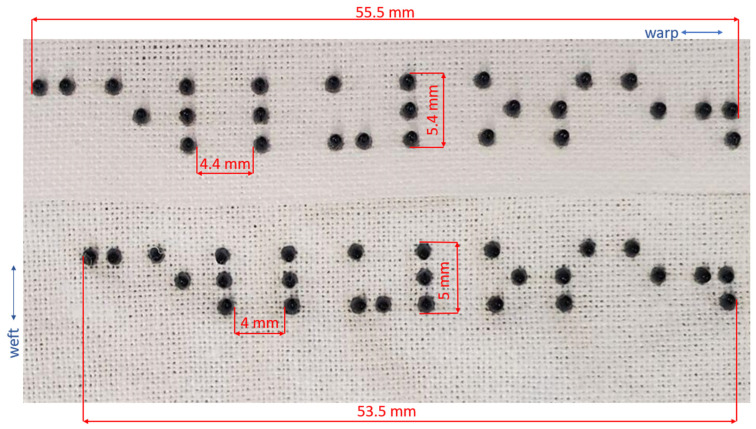
A close-up comparison of the resin dots on cotton before (**top**) and after (**bottom**) the accelerated washing.

**Figure 6 polymers-16-00486-f006:**
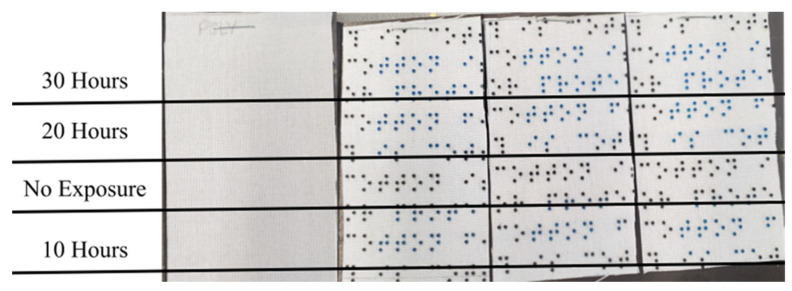
Polyester reference fabric and three samples after the lightfastness test.

**Figure 7 polymers-16-00486-f007:**
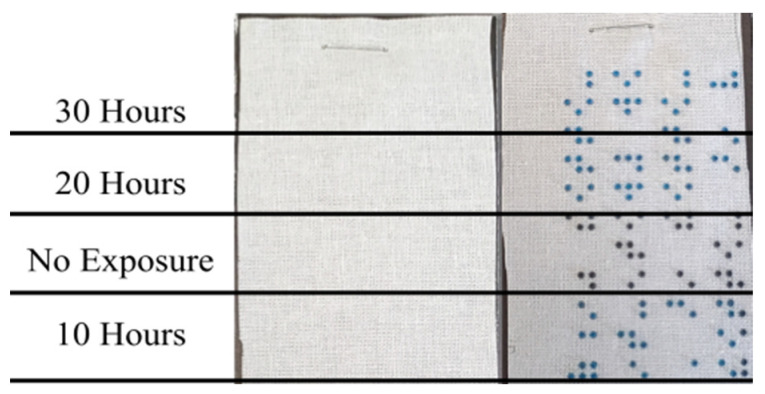
Cotton reference fabric and sample after lightfastness test.

**Figure 8 polymers-16-00486-f008:**
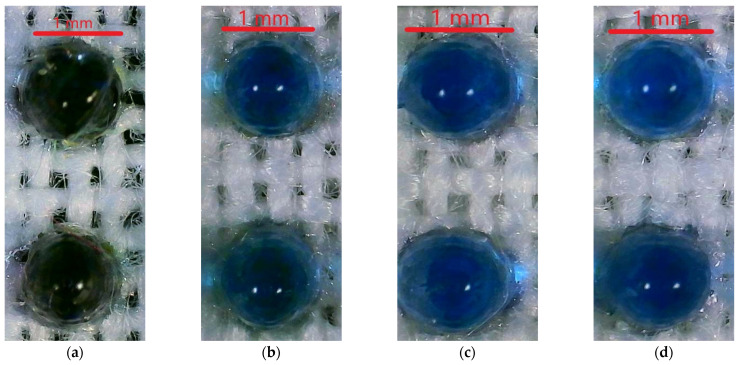
(**a**) Black dots unexposed to UV light; (**b**) darker blue dots exposed to 10 h of UV light; (**c**) blue dots exposed to 20 h of UV light; (**d**) blue dots exposed to 30 h of UV light. Images taken using Digital Microscope Model MX200-B, with an LED light source.

**Figure 9 polymers-16-00486-f009:**
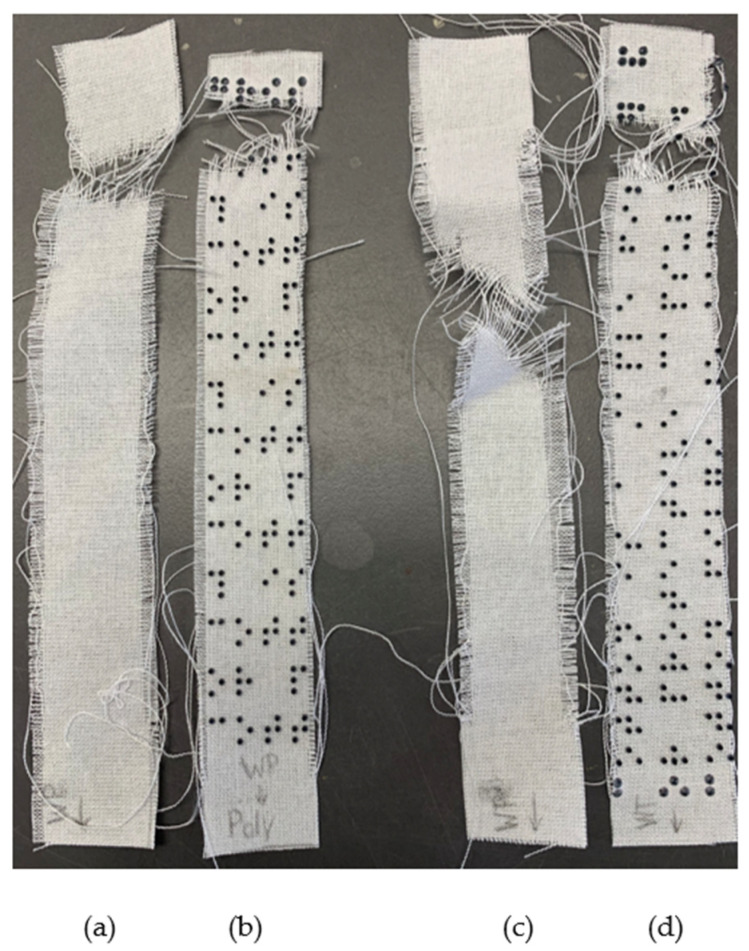
Polyester fabrics after the tensile test: (**a**) warp direction, reference; (**b**) warp direction, sample; (**c**) weft direction, reference; (**d**) weft direction, sample.

**Figure 10 polymers-16-00486-f010:**
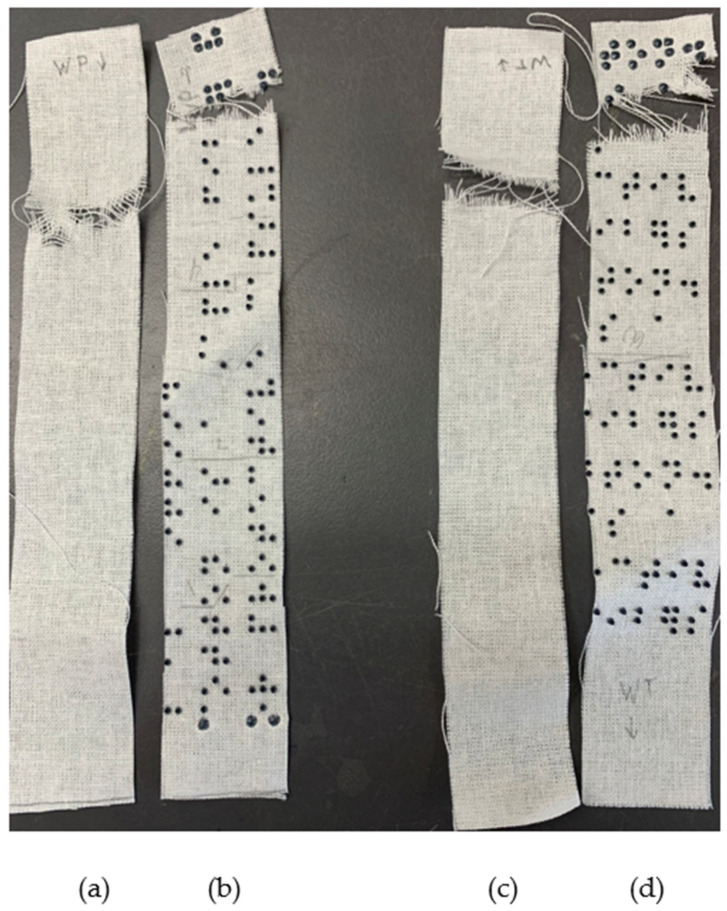
Cotton fabrics after the tensile test: (**a**) warp direction, reference; (**b**) warp direction, sample; (**c**) weft direction, reference; (**d**) weft direction, sample.

**Figure 11 polymers-16-00486-f011:**
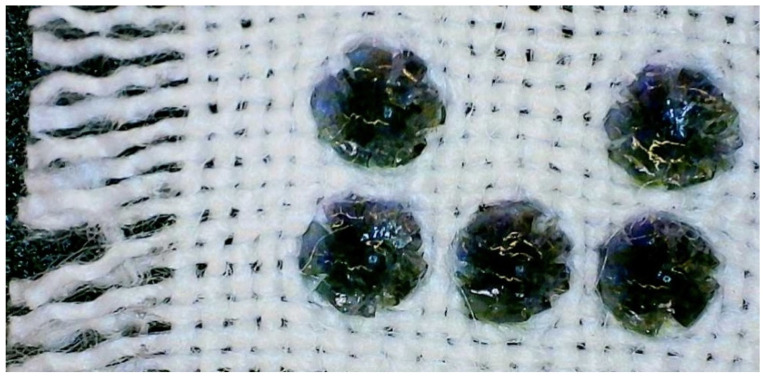
Digital microscope photo of crushed resin dots from the tensile test on polyester. Images taken using Digital Microscope Model MX200-B, with an LED light source.

**Figure 12 polymers-16-00486-f012:**
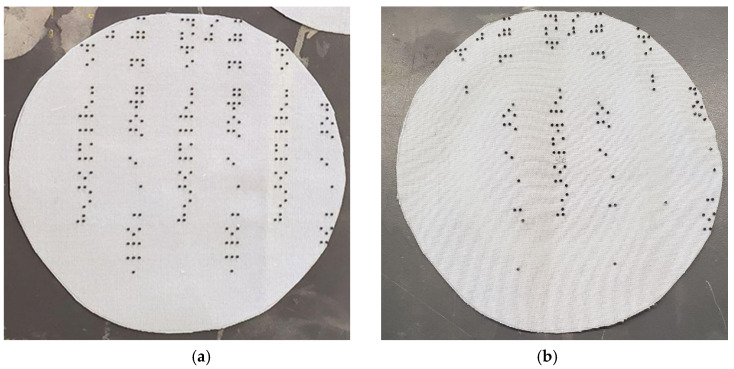
Abrasion resistance test on the polyester: (**a**) sample before the test; (**b**) sample after the test.

**Figure 13 polymers-16-00486-f013:**
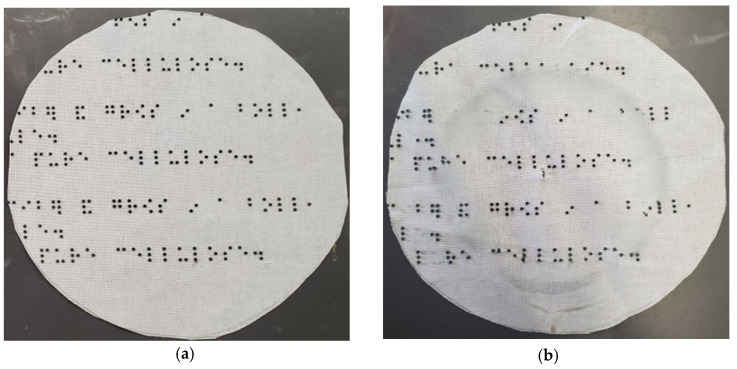
Abrasion resistance test on the cotton: (**a**) sample before the test; (**b**) sample after the test.

**Figure 14 polymers-16-00486-f014:**
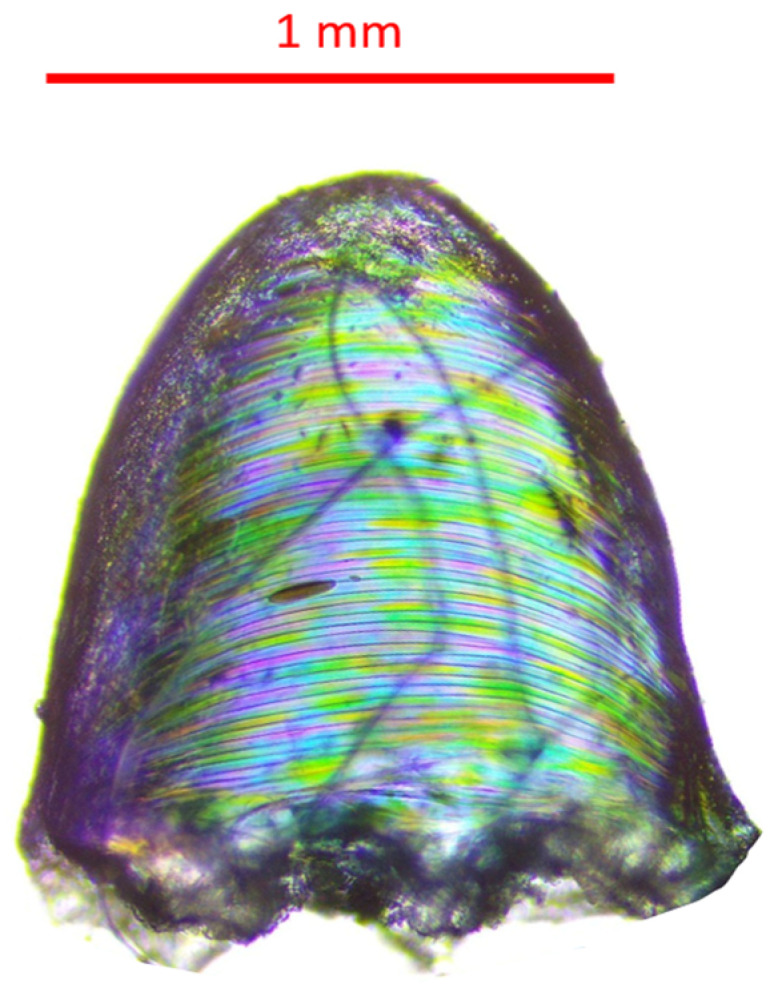
Microscopic image of a single dot observed in transparent light after being detached from 100% polyester fabric during the abrasion test. Microscope used: Nikon Eclipse E200.

**Figure 15 polymers-16-00486-f015:**
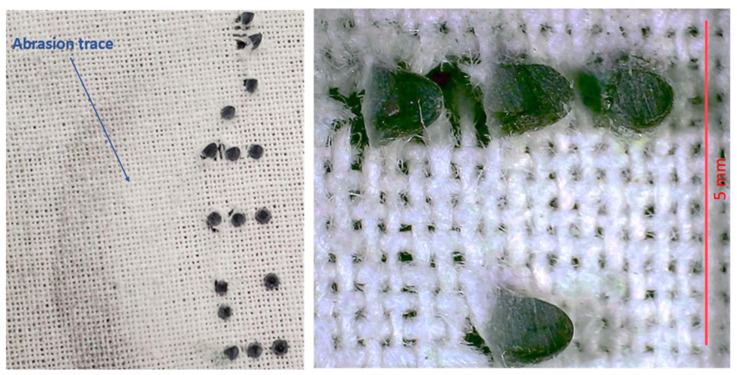
Close-up of the cotton sample after abrasion showing tearing around the resin dots.

**Table 1 polymers-16-00486-t001:** The characteristics of two fabric substrates, 100% cotton and 100% polyester (staple fibers), used in the 3DP process. The characteristics were measured on fabrics without 3DP.

Parameters	100% Polyester Substrate	100% Cotton Substrate
Weight [g/sq.m]	195	134
Thickness [mm] Warp Density [yarns/dm] Weft Density [yarns/dm]	0.0390 244 213	0.0330 256 228

**Table 2 polymers-16-00486-t002:** Breaking force and elongation averages for polyester references and samples.

	Polyester			
	Warp	Warp	Weft	Weft
Parameter	Reference	Sample	Reference	Sample
Breaking Force [lbs]	147.11	105.29	118.28	79.46
Elongation [%]	54.94	46.59	71.35	51.88
The average number of resin dots	0	154	0	158

**Table 3 polymers-16-00486-t003:** Breaking force and elongation averages for cotton references and samples.

	Cotton			
	Warp	Warp	Weft	Weft
Parameter	Reference	Sample	Reference	Sample
Breaking Force [lbs]	41.04	22.23	36.57	20.12
Elongation [%]	15.29	11.78	39.77	31.06
The average number of resin dots	0	134	0	127

## Data Availability

The research data are available in an Excel sheet file upon request.

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
