# Peer review of "Performance of Fabrics with 3D-Printed Photosensitive Acrylic Resin on the Surface"

_polymers, 2024, doi:10.3390/polym16040486_

Round 1

Reviewer 1 Report

Comments and Suggestions for Authors

1) there are no sections about materials and equipment, everything is listed in the text. It would probably be better to add.

2) Why for the tests in sections 2.4, 2.5, 2.6, one sample was used, shown with a pattern applied, and one control, while on crystals with polyester 3 with a pattern and 1 reference were used?

3) Results, section 3.1. Did the points themselves shrink? Has their acceleration changed after washing?

4) The photographs in Figure 8, 11, 14 were taken using what device? (said are just digital microscopic photographs) (a bit of a dubious comment, not necessarily included in the review)

5) Discussion, Section 4.1. They write that it is unclear how the dots affected the shrinkage of the fabric. Did comparison with the control sample show the same shrinkage?

Author Response

The replays to the questions and comments of Reviewer 1 are attached

Reviewer 2 Report

Comments and Suggestions for Authors

Thank you for sharing the work titles "Performance of fabrics with 3D printed photosensitive acrylic resin on the surface" with the journal of MDPI Polymers. The manuscript discusses cases of printing photosensitive acrylic resin on cotton and polyester fabrics. Both cases were able to pass colorfastness to washing tests with no transfer or color change, but they showed color change during colorfastness to light. Tensile tests showed a reduction in properties. Lastly, the adhesion of the 3D print was stronger in the cotton sample compared to the polyester and they were both easily removed from the fabric. Overall, its a very interesting application with a lot of ongoing challenges regarding mechanical properties and light sensitivities as mentioned in the text. 

I have the following suggestions/question to improve the work:

-Great work with the introduction and literature background describing the technology and its advancements.

-The uniqueness of the work is clear with adhesion being tested previously but the mechanical properties and changes due to usage is not well studied.

-The aim of the work and the outline of the tests studied is well described at the end of the introduction. Great job.

-Great goal in targeting visually impair people by using braille, I commend you for the usage of such application for 3D printing on fabric.

-Fig. 3 and Fig. 4 please clarify the strands shown by using letters such as (a) (b) etc.. rather than stating left and right.

-Colorfastness to light section (3.2): what is the reason for the color change? please clarify it, and is it just observed with UV light?

-Fig. 9 and all related figures please use letters to describe pictures of multiple samples to ease the understanding for the readers.

-Do you have any suggestions or guidance to researchers on how the 3D printed dots influence the shrinkage of cotton fabric?

-The color change may be influence by the combination of colors used. Have you tested each color mixed individually? or have you tried to recreate black using a different combination?

-Conclusion section: please state a summary followed by conclusions as bullet points to ease the understanding.

-Please add a recommendation section after the discussion and before the delivery of information.

-Please add a nomenclature at the end.

Great topic and good writeup, please work on answering my concerns.

Author Response

The replays to the questions and suggestions of the Reviewer 2 are attached.
